# Modelling Watershed and River Basin Processes in Cold Climate Regions: A Review

**Junye Wang** [1,*] , **Narayan Kumar Shrestha** [1] , **Mojtaba Aghajani Delavar** [1] , **Tesfa Worku Meshesha** [1]
**and Soumendra N. Bhanja** [2]

1   Faculty of Science and Technology, Athabasca University, 1 University Drive,
    Athabasca, AB T9S 3A3, Canada; shrestha.narayan@hotmail.com (N.K.S.);
    maghajanidelavar@athabascau.ca (M.A.D.); hopee2011@gmail.com (T.W.M.)
2   Interdisciplinary Centre for Water Research, Indian Institute of Science, CV Raman Rd, Bangalore,
    Karnataka 560012, India; soumendrabhanja@gmail.com
*   Correspondence: junyew@athabascau.ca; Tel.: +1-780-394-4883

**Abstract:** Watersheds in cold regions provide water, food, biodiversity and ecosystem service. However, the increasing demand for water resources and climate change challenge our ability to provide clean freshwater. Particularly, watersheds in cold regions are more sensitive to changing climate due to their glaciers' retreat and permafrost. This review revisits watershed system and processes. We analyze principles of watershed modelling and characteristics of watersheds in cold regions. Then, we show observed evidence of their impacts of cold processes on hydrological and biogeochemical processes and ecosystems, and review the watershed modeling and their applications in cold regions. Finally, we identify the knowledge gaps in modeling river basins according to model structures and representations of processes and point out research priorities in future model development.

**Keywords:** watershed modeling; cold climate; climate change; modeling of river basins

## 1. Introduction

River basins are multi-scale water-energy-land nexuses because they provide water and land and sustain life and therefore, are ecologically and economically significant for the development and sustainability of communities in many regions, such as Athabasca River Basin (ARB) [1,2], and the Elbow River watershed in Alberta, Canada [3]. However, industrial development and climate change in the river basins are affecting both the environmental and economic sustainability and the wellbeing of humans [1,4–8]. Particularly in cold regions, a river basin ecosystem is vulnerable to the climate change affecting permafrost thaw and glacier retreat, natural disaster, wildfire, oil extraction and deforestation [1,2,9,10]. Climate change can also cause a shift in peak stream flow in a river to winter and early spring due to the early melting of winter snow [11,12]. This can deviate from summer and autumn when demand is highest [4].

River basins include terrestrial and aquatic systems. Natural biogeochemical and hydrological processes interact with social and economic drivers through land use change and human activities at different scales. Social scientists and economists have different approaches to study land use and its change. Policy makers and social scientists have together identified the need to explore the potential indicators of how human activities and climate change affect land use change and associated impacts, such as sediment, water quality, greenhouse gas emissions, and toxic substances transferred by the agriculture and industry on the river, and hydrological and weather extremes. However, much remains unknown about how the impact of human activities and climate change influence the environment and society in river basins, such as groundwater and surface water resources, water quality, greenhouse gas emission, soil degradation and toxic substances transferred by the oil and gas industry to the river, and hydrological extremes. Against

this background, watershed models, as a system approach, are fundamental a system framework is to integrate the two natural biogeochemical and hydrological processes first, and then couple them with the social and economic processes for assessing all aspects of sustainable resource systems [13]. Such a scientifically based and widely accepted modeling framework would provide scientific evidence for trade-off, decision-making and policy in agriculture, industry, policy and the public.

Watershed models can be categorized into the stochastic and deterministic models upon their modeling approaches. Stochastic models are based on statistical regressions of the monitoring or experimental data represented by statistical distributions through selecting input parameters. The state-of-art stochastic watershed models are enhanced using Artificial Intelligence (AI), such as Genetic Algorithms (GAs) and Artificial Neural Networks (ANNs) [14]. However, this approach depends on data availability and empirical selections of parameters and is lack of mechanism representations. Deterministic models are process-based models, where mass, momentum, and energy conservations are formulated as a set of partial differential equations or water budget balances [13]. Here we focus on deterministic integrated watershed models.

An integrated watershed model is to incorporate greater levels of realism in evaluating how the biogeochemical, hydrological and social processes interact in river basins. In the past decades, many integrated watershed modeling frameworks have been developed to provide quantitative evidence for watershed management, such as the Soil Water Assessment Tool (SWAT) [15,16], the Variable Infiltration Capacity model (VIC) [17], the Hydrological Simulation Program—FORTRAN (HSPF) [18] and the Cold Regions Hydrological Model platform (CRHM) [19,20]. These models are distributed or semi-distributed by accounting for spatial variations in all variables and parameters and have been widely used for the optimal management and planning of water resources, while mitigating potential negative impacts on some ecosystems [21]. However, because of different focuses, success in implementing such model development depends on diverse knowledge and adequate tools, and financial and institutional resources. There is gap from a sensitive nature of a cold region to climate change, importance of process integration in watershed, watershed modeling to needs for an interdisciplinary review of watershed modeling. The gap might be key issues of watershed modeling for cold regions.

This review revisits watershed processes and system and analyze principles of watershed modeling. Next, we analyze characteristics of watersheds in cold regions and their impacts on hydrological and biogeochemical processes and ecosystems. Then we review the watershed modeling and applications. Finally, we identify the knowledge gaps in modeling river basins and research priorities in future model development provide a summary of the future research and developments of watershed models.

## 2. Watershed System and Processes in Cold Regions

### 2.1. Watershed System and Processes

A watershed is defined by natural topographic boundaries as an ecosystem of integrated terrestrial and aquatic systems, rather than the political boundaries, in which all of the incoming precipitation and snowmelt are collected into the stream reaches while a river basin is an area of land drained by a river, its tributaries and watersheds. The watershed function is generally defined as its response to the water entering its control volume [10,22]. The river shape, land topology, wetlands and riparian zones regulate the overland and subsurface flows as well as groundwater. Water budget in a watershed is balanced by precipitation, evapotranspiration, infiltration, and runoff. For a cold region, glacier, snowpack, permafrost, etc. (frozen water components) are also important parts of water budget. Some of the models discussed in this paper may not have a good process to capture those budgets. Watershed ecosystems are controlled by a suite of hierarchically nested physical, chemical, and biological processes operating over space and time.

A watershed consists of terrestrial or aquatic sub-ecosystems, including forest, grasslands, arable, urban, and wetlands. Water, energy, air, vegetation and land interact within

the watershed. Some main watershed processes can be broken down into specific functions and characteristics, including hydrological processes (i.e., overland flow, evaporation, infiltration, groundwater recharge, and erosion), biogeochemical processes (i.e., nutrient cycling, pollution transport and fate, mineralization and organic matter decomposition), and land-surface interaction (i.e., plant growth, photosynthesis, and evapotranspiration). In cold regions, there are four special elements, including snow, permafrost, freeze-thaw cycle and glaciers as well as lake and river ices (Figure 1) [12]. The biogeochemical processes, such as nitrification, denitrification and decomposition, can be affected by soil water, temperature and pH, depending on various hydrological processes. Human activities, such as land use changes, mining and water policy can also affect water balance and biogeochemical processes. Intensive agricultural practices, such as grazing, irrigation and fertilization, can affect hydrological and biogeochemical processes, and cause land degradation, such as salinization and desertification [21].

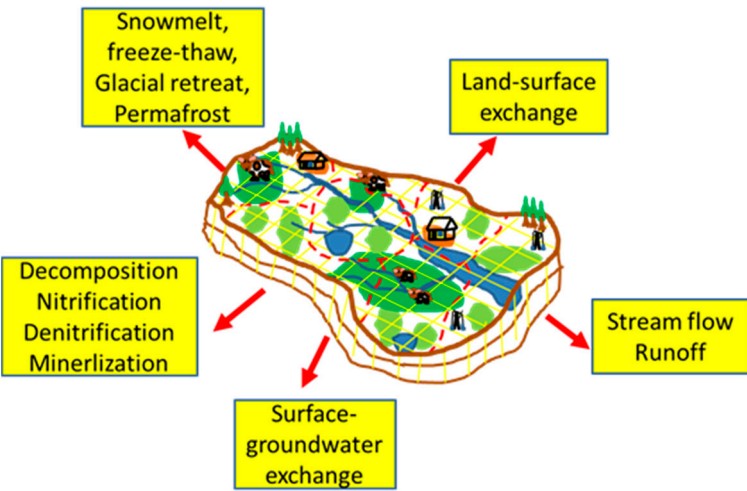

**Figure 1.** Watershed ecosystem processes at larger spatial scales (heavy arrows).

### 2.2. Principles of Watershed Modelling

Watershed modeling is to quantify hydrology and biogeochemistry with associated ecosystem functions, such as plant dynamics, in the watershed. An ideal basin-oriented strategy is to simulate the integration of watershed hydrology, nutrient, and sediment reactive transport below and above ground [23]. However, in reality, we cannot include all detailed watershed processes. Hence, abstraction through parameterization and simplification is necessary in representing complexity level of watershed system and processes. The parameterization and simplification are a compromise between reality, feasibility, and simplicity in a watershed model and are to use some replacement of watershed processes using a model of similar but simpler, more empirical conceptual structure. The point is not a perfect solution but a replacement to reflect the physical reality as good as possible. Thus, a watershed model is an approximation of its reality to describe the watershed processes and system which can retain most of its important characteristics [13,24].

Conceptually, it is quite natural to use a symbolic watershed model to represent some of the natural hydrological and biogeochemical processes of the real system according to the first principle. A set of equations of mass, momentum, and energy conservation along with initial and boundary conditions can be established to describe the streamflow, infiltration, subsurface flow, and baseflow in a watershed system driven by the weather conditions, such as precipitation and solar radiation, soil and plants. Unfortunately, these equations cannot be solved analytically and can be solved only for very simple geometries or boundary conditions. Therefore, a watershed domain is usually distributed on discrete grids and the partial differential equations are replaced by their discrete equations. For example, the Richards equation of water transport in soil are solved by using finite difference

method [25–29]. However, such models require extensive data and physical parameter, which may not be efficient to set up the model. Furthermore, a watershed model needs to consider readily available data with sufficient efficiency of computer power and memory while simulating large river basins. This will allow realistic simulation of some medium complexity, which generally includes relatively detailed data inputs such as topography, land use, soil, weather, and water quality. Thus, many distributed models delineate a watershed into a multi-level hierarchically distributed drainage system using reasonable spatial discrete elements, such as sub-basins and hydrological response units (HRUs) in SWAT [30], CRHM [19] and HSPF [31], and a grid of large (>>1 km), flat, uniform cells in the VIC (Table 1). These watershed models not only represent the fundamental water and sediment transport processes, but also combine the empirical and statistical equations such as nitrification, denitrification and decomposition. The use of the lumped parameters and HRUs or similar units are to reduce the grid number because the computational cost depends on the numbers of grid cells and parameters [32].

**Table 1.** Characteristics of watershed models in cold regions.

| Model Name | Spatial Resolution | Time Step | Snowpack Model | Snowpack Layer | Water Quality | GHGs | References |
|---|---|---|---|---|---|---|---|
| HSPF | HRUs | Sub-daily | Snowmelt/sublimation/ compaction/albedo/ blowing/radiation/interception | Two layers | Yes | | [18,33,34] |
| CRHM | HRUs | Daily | Snowmelt/sublimation/ compaction/albedo/ blowing/radiation/interception | Two layers | | | [19,35] |
| CRHM+ WINTRA | HRUs | Daily | Snowmelt/sublimation/ compaction/albedo/ blowing/radiation/interception | Two layers | Yes | | [36,37] |
| SWAT | HRUs | Daily | Snowmelt/sublimation | One layer | Yes | | [15,38–41] |
| VIC | Grid cell (>>1 km$^2$) | Daily | Snowmelt/sublimation/ compaction/albedo/ blowing/radiation/interception | Two layers | Yes | | [42,43] |
| SWAT-DayCent | HRUs | Daily | Snowmelt/sublimation | One layer | Yes | $N_2O/CO_2$ | [44–48] |
| SWAT-MKT | HRUs | Daily | Snowmelt/sublimation | One layer | Yes | $N_2O/CO_2/NEE$ | [49–51] |
| VIC-CropSyst | Grid cell (>>1 km$^2$) | Daily | Snowmelt/sublimation/ compaction/albedo/ blowing/radiation/interception | Two layers | Yes | | [52–54] |

In every HRU unit, the parameters of a watershed model represent a volume-average of their variability over the unit that consists of homogeneous land use, management and soil properties, slopes and weather conditions. In the HRUs, only percentages of the land uses are represented, and parameters, such as soil properties and land uses are not identified spatially in these models [30]. For example, in the VIC model, the sub-grid variability is handled in terms of statistical properties [30,42,55]. Therefore, land-atmosphere fluxes, and the water and energy balances are calculated at a daily or sub-daily time step but horizontal flow between HRUs is not considered [30,42]. The water balance is calculated among rainfall, runoff, infiltration, evapotranspiration, storage, snowmelt water, lateral subsurface flow from the soil layers and baseflow at each HRU. Flows are aggregated from all HRUs to the sub-basin level, and then directed to stream reaches along the directions of element slopes using either the variable-rate storage method or the Muskingum method [19,56]. Sediment, nutrient, pollutants, and bacteria loading from each HRU are also collected at the subbasin outlets, and are directed to channels, ponds, wetlands, and lakes. Contributions from point sources and urban areas can be inputted from HRUs and be exported from each sub-basin.

## 3. Characteristics of Watersheds in Cold Regions

In cold regions, there are four special processes related snow and ice, including snowpack and snowmelt, permafrost, freeze-thaw cycle, glaciers and ices in lakes and rivers (Figure 2) [12]. These processes can affect hydrological and biogeochemical processes and ecosystems. Watershed models can not only consider a wide range of watershed-scale hydrologic cycle, such as stream flow, infiltration, depression storage, evaporations, and baseflow, but also these water issues in cold regions.

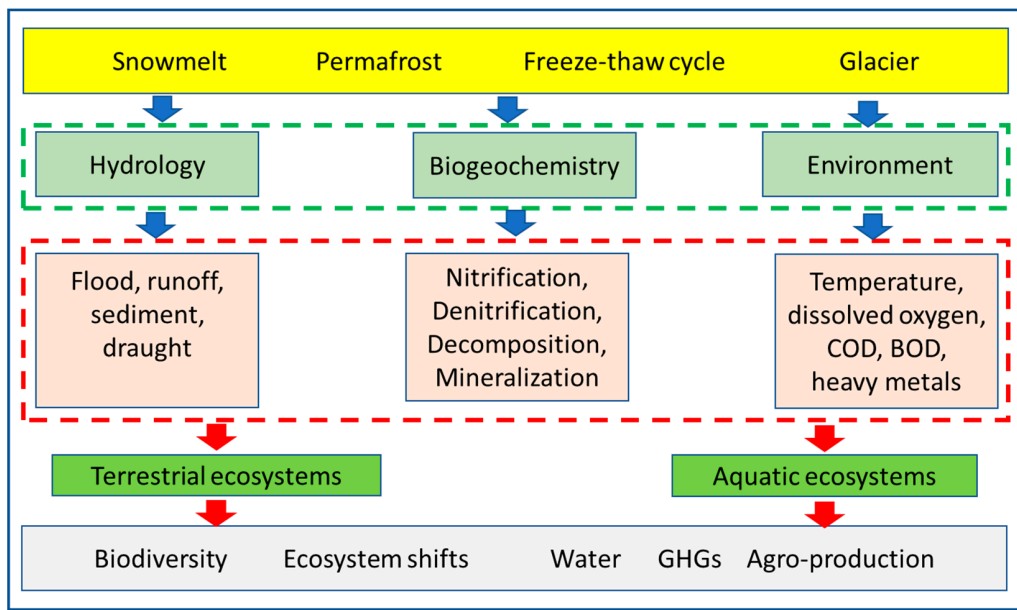

**Figure 2.** Effects of snow and ice processes on hydrology, biogeochemistry, and ecosystems.

### 3.1. Impacts on Hydrological Processes

Alpine glaciers are the most important resource of freshwater as sources of headwaters in many river basins. For example, the Gangotri Glacier in the Himalayan Mountains is the source of the Ganges River [57] and Columbia Glacier is the source of the Athabasca River basin, Canada [1]. Artic and semi-arctic regions are covered by snowpack and ice in more than 6 months in every year. Unlike alpine glaciers, they are not limited to mountainous areas. Once snow and ice melt, melting water of snow and ice contributes to surface and ground water sources, which are routed to the river and reservoirs. If global warming, glaciers and snowpack can melt more quickly than their usual. This can significantly change the exchange of water and heat between soil surface and the air [1,58]. Cai et al. [59] found the shorter ice cover durations for 40 lakes and the longer duration for 18 lakes in the Tibetan Plateau during the period from 2000–2017. Su et al. [60] studied the effects of snowmelt on suspended sediment and phosphorus in two watersheds (Bras d'Henri watershed and Bras d'Henri watershed) in eastern Canada. They showed that soil frozen status and rainfall substantially affected suspended sediment and phosphorus runoff during snowmelt.

Unlike glaciers and snow, permafrost and freeze-thaw cycles are a subsurface phenomenon that cannot easily be observed remotely. All soils in cold regions are subjected to seasonal freeze-thaw cycle. When temperature is higher than 0 °C, soil water freezes and forms a thin layer of ice. When temperature is lower than 0 °C, soil water thaws. In Edmonton region, Canada, there are about 91 freeze-thaw cycles in the range of 80–100 per year for the last 90 years on an average [61]. However, distribution and change of freeze-thaw cycles in many mountainous regions are less understood than glaciers, permafrost thaw, or snow, and it can only be inferred [62]. Ices in rivers and lakes are a typical of physical phenomena in cold regions. In Canada, Rokaya et al. [63] found highly variable trends in the timing and magnitude of ice-jam floods in the northern hemisphere and

rivers in the mountains. They found that patterns in the timing of ice-jam floods tended toward to earlier times in southeastern and western Canada whereas Atlantic Canada was experiencing delayed timing.

### 3.2. Impacts on Biogeochemical Processes

The freeze-thaw cycles, permafrost, glacier retreat and change of snowpack change the exchange of water and heat between soil surface and the air duration. This can impact biodiversity, biogeochemical processes, carbon storage and greenhouse gas emissions (GHGs) [8]. Therefore, cold regions are vulnerable to global warning. Many studies have been performed to study effects of snowpack, permafrost and freeze-thaw cycles on nitrate ($NO_3^-$) and carbon (C) and carbon dioxide ($CO_2$), nitrogen dioxide ($N_2O$) and methane ($CH_4$) emissions using closed chamber technique, eddy covariance, IPCC inventory and modeling [58,64–73].

Climate change can increase soil temperature, leading to the snow cover reduction during the spring season and permafrost [74]. Soil temperature and soil freezing can affect the nitrification, denitrification and decomposition [65]. It has been reported that soil freeze-thaw cycles change the structure and activity of microbial activities and denitrifying enzymes, leading to increase of $N_2O$ emissions [58,75,76]. Ruan and Robertson [77] showed that the freeze-thaw cycles and reduced snow cover have a substantial effect on soil carbon and nitrogen cycles, leading to $N_2O$ emissions. Peatlands in cold regions stored a vast pool of immobile C and N in permafrost [78–80]. Thawing permafrost can cause decomposition and remobilization of the stored carbon due to global warming up and trigger greenhouse gas emissions of $CO_2$ and $CH_4$ [81–83]. Olefeldt and Roulet [84] investigated carbon stock in thermokarst landscapes. They found that thermokarst landscapes included about 50% of the soil organic carbon within the study region. Burd et al. [85] showed that derived dissolved organic carbon (DOC) of peatlands is important sources to creek and river systems. They found seasonal shifts in export of DOC and nutrients in burned and unburned peatland-rich watershed, Northwest Territories, Canada.

### 3.3. Impacts on Water Quality and Aquatic Ecosystem

The glacier retreat can cause ecosystem and environmental changes, such as the amount and timing change of freshwater available for plants and animals that need fresh water to survive [86]. Organisms, such as many types of microorganism, depend on temperature and dissolved oxygen water for survival. Some fishes may not be able to fit to a temperature and chemicals in river and lakes. Modelling of ecosystems related to snowmelt, freeze–thaw events is still a challenge due its complex interactions among organisms and hydrological processes. Sharma et al. [87] studied surface water quality in the Gangetic Plain from Himalayan glaciers. They found that melting Himalayan glaciers could contribute to polychlorinated biphenyls (PCBs) and polycyclic aromatic hydrocarbons (PAHs) because of accelerating the release of stored anthropogenic legacy pollutants. PCBs have been also detected in runoff from melting glacier in the central Tibetan [88] and in remediate soils in Alberta, Canada [73].

### 3.4. Modelling of Snow and Soil Ice Processes

Most of existing watershed models use one-dimensional structure to simulate vertical mass and energy fluxes in the snowpack. Generally, the snowpack can be divided in a single-layer [58,89,90] or multilayers [91,92]. A detailed multilayer snowpack model requires several combinations of different snow process representations to simulate mass and energy conservation in the snowpack [93]. For example, snow-albedo feedback strongly depends on Snow Cover Fraction (SCF) [56,94]. SCF is not available in most situations and requires to be formulized. Therefore, explicit representation of snowpack processes in the existing watershed models was often simplified not to mimic real-world layering. Instead, they are numerical constructs as one-layer snowpack in SWAT [56], and two-layer snowpack in CRHM [95] and VIC [96]. Snow processes are parametrized for many

processes, such heterogeneous distribution of temperature in snowpack, snow and canopy gap influence on radiation and albedo, wind blowing snow, and snow cover fraction. Snowpack thickness in one layer model is simplified to one layer to calculate the snow and water equivalent balance between snowfall, snowmelt and sublimation [56]. Similarly, in the two-layer model, ground snowpack accumulation and ablation are calculated from the balance of snow and water equivalent of snow between snowfall, snowmelt and sublimation. Snowmelt is calculated as a function of the air and snowpack temperature, the melting rate, latent heat flux and the areal cover of snow. The density of snowpack will tend to increase with age due to compaction and decrease due to new snowfall. The rate of compaction would be calculated in term of the actual change in the depth due to compaction in HSPF [18]. The dynamic density and thickness between layers of snowpack were updated daily. The energy and mass balances between the snowpack layers are calculated with coupled snow processes, such as the accumulation, storage, and melt of snow [97,98]. Water balance was simulated considering atmospheric input (rainfall and snowmelt), output (evaporation and transpiration), and transport among soil layers. Heat transfer between layers was calculated from the volume-weighted heat capacities of their constituents and temperature gradient. In the soil layers, soil temperature dynamics were simulated by solving the one-dimensional heat conduction equation. The upper boundary condition is the ground surface or snow surface. The geothermal heat flux is determined by the surface energy balance and the lower boundary condition. Soil thawing and freezing, and the associated changes in each soil layer, are calculated as ice and liquid water fractions using heat balance [91].

## 4. Watershed Modelling and Applications

Generally, watershed modeling is used to simulate hydrological processes, such as stream flows, sediment, runoff, and baseflow. In cold regions, the models require considering physical snow and ice processes, such as snow redistribution, sublimation, snowmelt, freeze–thaw cycles, infiltration into frozen soils, and permafrost [19]. The snowmelt algorithms within SWAT have been developed in mountainous basins [97,99]. The advanced watershed modeling includes biogeochemical processes, such as nitrification, denitrification, decomposition and other biogeochemical cycle component transport and fate [49]. Furthermore, some watershed models also incorporate some ecosystem and ecological processes, such as aquatic ecosystems [40].

### 4.1. Stream Flow, Freshwater Resources and Sediments

Freshwater resources are important for the sustainability of ecosystems and for the socio-economic development in river basins. The snowmelt water plays an important role to maintain streamflow in the cold river basin. The winter flows are low due to freezing and the spring flows are high due to snowmelt. Therefore, a climate change impacts on stream flows in a cold region should be considered. Shrestha et al. [1] investigated climate change impacts on freshwater resources the Athabasca River Basin (ARB) in Canada. They used SWAT to compare two Representative Concentration Pathway (RCP) scenarios (RCP 4.5 and 8.5). Their results showed significant increases (16–54%) in annual streamflow which poses flooding problems across the basin. Leong and Donner [100] studied the response of streamflow in the ARB under climate change using the Integrated Biosphere Simulator (IBIS) and the Terrestrial Hydrology Model with Biogeochemistry (THMB). They found that future climate change increases streamflow in winter while decreases streamflow in summer. Furthermore, the frequency of summer low streamflow is projected to an increase by up to 85% in the highest emissions scenario in the end of the century. Jiang et al. [101] used the ArcSWAT model to evaluate the impacts of climate change on seasonal and annual streamflow in the Nicolet River watershed, Southern Quebec. They found that streamflow peak was expected to an increase of 13% annually. An increasing trend of earlier snowmelt could cause the shift of the peak streamflow to the late winter and earlier spring. Bajracharya et al. [102] evaluated effect of climate change on water

balance in the Kaligandaki Basin, Nepal under two Scenarios (RCP 4.5 and RCP 8.5) using SWAT model. They indicated that snowmelt contribution will largely be affected by climate change, and it is projected to increase by 90% by 2090. Zhou et al. [103] used CRHM to evaluate streamflow in the west of China. They found that snowmelt contributed to runoff in the high alpine Binggou basin, while soil freezing/thawing did a runoff in the steppe Zuomaokong. The seasonal snow sublimation loss contributed to 47% of 291 mm water equivalent of snowfall in Binggou basin. Dibike et al. [104] simulated the Athabasca watershed snow response to a changing climate under two scenarios RCP 4.5 and RCP 8.5 emissions scenarios using the VIC model. Their results showed SWEmax increase in the middle sub-basin of the ARB for RCP 4.5, while it decreases throughout the ARB for RCP 8.5. Eum et al. [43,105,106] analyzed effects and uncertainties of different gridded climate data on hydrological responses using the VIC model in the ARB. Faramarzi et al. [107] analyzed uncertainty of dynamic freshwater scarcity in semi-arid watersheds of Alberta, Canada. They found severe blue water scarcity in spring and summer months due to irrigated agriculture. In contrast, in winter months the blue water scarcity was mostly due to the demands of petroleum or other industries. Shrestha et al. [108] simulated climate-induced hydrologic changes in the Lake Winnipeg, Manitoba, Canada, watershed using SWAT model. They found that the change of future precipitation and temperature results in higher total runoff, earlier snowmelt, and discharge peaks. Morales-Marin et al. [109] simulated the timing of river freeze-up and ice-cover breakup in the ARB using a physically-based semi-distributed hydrological model. They found that in the main tributaries of the ARB, freeze-up timing shifts from the last week of September to the second week of November. The breakup timing shifts from the second week of March to the last week of May. Krogh and Pomeroy [110] simulated stream flow and vegetation in an Arctic basin using CRHM. They found that stream flows were influenced by decreasing snowfall, deepening active layer thickness, earlier snow cover depletion and diminishing annual sublimation during 1960–2016.

Wu et al. [111,112] simulated effects of snowmelt on soil erosion in a mid-high latitude upland watershed using SWAT. They evaluated soil erosion characteristics in response to temperature and precipitation in a freeze-thaw watershed in the Abujiao River Basin, Heilongjiang Province, China. They found that seasonal variations in soil erosion was strictly correlated with surface runoff and snowmelt. Snowmelt runoff played a substantial role in the soil erosion process in the snowmelt period. Low temperature with adequate precipitation might lead to higher soil loss in the freeze-thaw areas. Shrestha and Wang [2] simulated effects of climate change on sediment and erosion under in the ARB. They examined channel erosion and deposition under two scenarios (RCP 4.5 and 8.5) using SWAT. They found that the spatiotemporal variability is closely followed with the trend of streamflow and more than two-fold increases of sediment load are projected. Future increases on sediment loading are projected up to 0.94 t/ha/year in the agricultural lands.

*4.2. Water Temperature, Water Quality and Aquatic Ecosystem*

Glacier retreat and snowmelt can affect water temperature, water quality and aquatic ecosystem [113]. Water temperature is an important indicator of energy transfer processes at the water-atmospheric interface, which reflects the effects of air temperature, solar radiation, snowmelt, permafrost, wind speed and streamflow conditions on the heat transfer process [114]. Du et al. [114] developed an equilibrium stream temperature model within SWAT framework to study the effects of stream temperature on the chemical reaction rates in the aquatic environment in the ARB, Canada. They found that the chemical reaction rates and concentrations changed in magnitude when water temperature changed. Du et al. [115] simulated effects of climate change on stream temperature and stream ecosystem in the ARB. Their results showed that annual stream temperatures could increase 0.8 to 1.1 °C in mid-century and 1.6 to 3.1 °C in late century predicted by three different climate models. Increasing stream temperatures can influence dissolved oxygen concentrations and increasing the biochemical reaction rates in the aquatic system. This indicated that

stream temperature simulation is important for water quality modeling for biodiversity and sustainability in aquatic ecosystems. Shrestha and Wang [38] improved air-water oxygen exchange and the re-aeration process of the SWAT model to simulate impacts of climate change on water quality in the ARB under two emission scenarios (RCP 4.5 and 8.5). Their results showed that the modified SWAT could improve accuracy of the water temperature and dissolved oxygen in the cold regions. They indicated that the future pollutant and nutrient concentrations could be reduced about 50% of carbonaceous biochemical oxygen demand (cBOD), 20% of total nitrogen, and 60% of total phosphorus. Tang et al. [116] simulated hydrodynamics and water quality for rivers in the Northern Cold Region of China using a two-dimensional water quality model (i.e., Environmental Fluid Dynamic Codes). They found that the decay rates of chemical oxygen demand Chromium ($COD_{Cr}$) and $NH_3N$ were 0.03/day and 0.05/day respectively in the Mudan River in the northeastern region of China, during the open-water months while they were 0.01/day and 0.02/day during the ice-covered months. Mekonnen et al. [117] used a modified SWAT model to simulate nutrient export in the Assiniboine River watershed, Saskatchewan, Canada. They found that cover crops and filter strips reduce nutrient export from the study watershed. Shakibaeinia et al. [118] simulated dissolved oxygen (DO) and nutrients (i.e., nitrogen and phosphorus) using 1D and 2D hydrodynamic and water quality models in the lower Athabasca River (LAR) ecosystem, northern Alberta, Canada. They externally coupled with the 1D river ice process models to account for the cold season effects. They found that the winter ice cover could reduce the dissolved oxygen concentration and a fluctuating temporal trend during summer. There are substantial differences in concentration between the main channel and flood plains. Meshesha et al. [86] simulated spatiotemporal patterns of water quality using a modified SWAT model and analyzed the impacts of water quality on aquatic ecosystem in the cold climate region, in Alberta, Canada. They found that the concentration of DO reduced as the water temperature increased and volume of stream decreased. Costa et al. [37] coupled a process-based N model, WINTRA, with CRHM to simulate nutrient cycle in an agricultural watershed in Manitoba, Canada. Their results showed that Total Dissolved Phosphorus (TDP) peaks generally were earlier than nitrate peaks but reduced faster afterwards. This implicated that plant residue TDP contributed to early snowmelt run-off.

Pollutant transport models have been developed within the SWAT model to simulate PAHs from oil sand regions in the Muskeg River, Alberta, Canada, watershed of the Athabasca oil sands region [40] and heavy metals [39]. They showed that the rainfall-runoff event in spring and summer is the main drivers for the hot moment of PAHs and heavy metal transport in the rivers. Saari et al. [119] predicted iron transport in boreal agriculture-dominated watersheds under a changing climate in the River Mustijoki, Finland, using the SWAT model. They found that high iron transport period is already spanning from spring snowmelt period to autumn and winter. This change is likely to increase in coming decades.

### 4.3. Biogeochemical Processes and Greenhouse Gas Emissions

Cold climate regions are one of the most sensitive to climate change and human activities in the world. This can induce greenhouse gas emission, due to permafrost, snowmelt and freeze-thaw cycles [8,12]. However, the widely used watershed models were traditionally developed to investigate hydrological processes, such as runoff, infiltration, and sediments [42,55]. Biogeochemical components in watershed models like SWAT and VIC require to be improved by incorporating more realistic, process-oriented parameterizations of plant-growth and soil microbially-mediated for simulating plants growth and soil nutrient turn over [21]. There are three approaches to improve biogeochemical models: (1) single input and output exchange or share data through database or files, (2) embedded code as a subroutine through computer memory, and (3) integrated or merged codes [21]. Ghimire et al. [120] performed a review of recent progresses in modeling $N_2O$ emissions using SWAT model. Here, we review some remarkable progresses interdisciplinarity between biogeochemistry and hydrology in watershed modeling in

cold regions. Shrestha et al. [121] integrated the nitrification and denitrification model of agro-ecosystem model, Daycent, into SWAT to simulate from grazing and crops. Shrestha and Wang [46] simulated $N_2O$ emission in the ARB, Canada in current and future climate. Melaku et al. [48] incorporated soil respiration model of Denitrification-Decomposition (DNDC) into SWAT model to simulate effect of snowmelt on $CO_2$ emissions in the ARB. A microbially mediated kinetics and thermodynamic (MKT) model has been developed by Bhanja et al. [50,51] within the SWAT model. The model has successfully simulated $N_2O$ and $CO_2$ emissions in the ARB. They incorporated soil respiration model into SWAT model to $CO_2$ emissions and net ecosystem exchange (NEE) in the ARB [49].

*4.4. Wetlands, Groundwater, and Bacteria Transport and Fate*

Groundwater recharge is strongly influenced by snowmelt [122–124]. The groundwater modules in the watershed models are limited due to lack of representation of water table and the belowground morphologies although some small watersheds have been modeled [30]. Therefore, some watershed models coupled groundwater models based on Richards' equations, such as SWAT-MODFLOW [125,126]. Aliyari et al. [127] presented a model based on SWAT-MODFLOW to link MODFLOW pumping to groundwater irrigation in South Platte River Basin, Colorado, USA. Their results showed that this model could be used for large agro-urban river basins to assess water resources supply under different scenarios. A coupled model of SWAT-MODFLOW was based on the hydrologic response unit (HRU) in which SWAT exchanges data with the finite-difference grid cells in MODFLOW. Thus, the coupled SWAT-MODFLOW did not simultaneously simulate interaction between surface and subsurface flow. The integrated model requires nearly all the individual input files for each of the two original models: MODEFLOW and SWAT, to run the integrated SWAT-MODFLOW [21]. Melaku and Wang [128] modified groundwater-module in SWAT model to represent two-way groundwater-surface water interactions through considering evapotranspiration at both locations (Lethbridge and Barons). The results showed that the model captured daily water table dynamics, and the water table fluctuation.

Wetlands are an interface between the terrestrial and the aquatic systems in river basins. They could buffer floods, improve water quality and reduce sediment in river systems. The wetland submodels in the current watershed models like SWAT is insufficient to represent snowmelt processes, soil heat transfer, groundwater table and biogeochemical processes in cold river basins [55]. Melaku et al. [48] modified the SWAT model to improve modules of snow, soil temperature and $CO_2$. Then their model was used for predicting snow depth, soil temperature at different depths and carbon dioxide emission from wetlands in the ARB, Canada. Perez-Valdivia et al. [129] simulated impacts of wetland drainage on hydrology using the SWAT model in the pipestone creek watershed, Canada. Their results showed that drainage flow could increase by about 50, 79 and 113%, respectively in spring for the 1997–2009 period. In contrast, annual flow volumes increased by about 43, 68, and 98%. Stone et al. [130] simulated wetland discharge due to permafrost thaw-induced land cover change in the Scotty Creek, Northwest Territories, using the CRHM for the period of 2009 to 2015. They found that total annual discharge reduced by 2.5% while every 10% decrease in permafrost area. This is because the peatland thaw increased surface storage capacity and landscape evapotranspiration, and reduced discharge. The peatland thaw also reduced seasonal surface runoff and peak discharge due to increases in the flow path routing. Muhammad et al. [131] simulated streamflow of the Prairie Pothole Region (PPR), Canada using SWAT. They found that changing climate substantially increases streamflow (~200%) in winter while decreases streamflow (~11%) in summer. Bohn et al. (2013) simulated the effects of surface moisture heterogeneity on wetland $CH_4$ emissions in the West Siberian Lowland using the VIC. They found that effects of thawing permafrost on $CH_4$ fluxes may have been overestimated in the permafrost zone.

Bacteria in rivers can originate from a variety of point and/or not-point sources in a watershed. [132] simulated waterborne fecal coliform, and *Escherichia coli* in Payne River watershed in eastern Ontario, Canada, using SWAT. They showed that the SWAT did

not accurately capture bacteria transport and was not sensitive to an observed reduction in the cattle population. Shrestha et al. [133] simulated *Escherichia coli* dynamics in the river Zenne (Belgium) using the SWAT model. Their results showed that the bathing water, with occasional combined sewer overflows from the Brussels' sewer systems caused the water pollution. *Escherichia coli* loading increased above European standard in the river. Meshesha et al. [41] incorporated a pH module in SWAT model to simulate transport and fate of *Escherichia coli* and fecal coliforms in the ARB. They found that high loads of fecal coliforms were 34 CFU/100 mL in summer and 24 CFU/100 mL in spring respectively. In contrast, lower loads of fecal coliforms were 0.65 CFU/100 mL in autumn and 2 CFU/100 mL in winter respectively. Hwang et al. [134] simulated cumulative effect of wastewater discharge on acesulfame and *Escherichia coli* using an integrated surface/subsurface hydrologic model in Grand River Watershed (GRW) in Ontario, Canada. They found that simulated transient *Escherichia coli* levels downstream of wastewater in the watershed were significantly lower than observed values. This was explained that agricultural sources of *Escherichia coli* might significantly contribute to the watershed.

## 5. Future Research

The wide use of the widely used watershed models, such as SWAT and VIC, shows their promising in simulating watershed because of their outstanding advantages of intermediate complexity for environmental assessment. Despite many successes, the complexity of watershed modeling depends on the temporal and spatial resolution, representations of watershed processes, and availability of extensive data, such as climate, soil type, land use, land cover, elevation, and geologic [135]. Particularly, watershed processes in cold regions are mainly featured by spring snowmelt of the winter snowpack, permafrost, and freeze-thaw cycles. There are clear knowledge gaps in the distribution and characteristics of cryospheric variables in the extent and ice content of permafrost in mountains, current glacier ice volumes, trends in lake and river ice, and the spatial and temporal variation of snow cover [12,136]. Particularly, water and heat transport in snowpack layers are a three-dimensional process. In this regard, the widely used watershed models, both one-layer and two-layer models, consider only vertical water transport and snow accumulation. Ideally, a watershed model should consider river morphology, biogeochemistry, surface water and groundwater interaction, climate, and land-surface interactions. However, most of currently typical watershed models use widely spatial discrete elements, such as sub-basins and HRUs in the SWAT and the coarse grids in the VIC. While this made a watershed model simple, the HRUs are not landscape dependent [55]. A simple fact is that the spatial scales of the watershed processes involved range over many orders of magnitude from the microbial processes at soil pore scale to the stream flow at regional scale of many thousands of square kilometers [21,137]. Although water and nutrient can be calculated at every grid cell or HRU, it is lack of mass, energy, and momentum conservations according to the first principle to describe important physical processes, such as groundwater and surface water interaction, and biogeochemical processes. There are three main problems: (1) lateral transport of water and nutrient between grids, (2) representation of hydrological and biogeochemical processes, and (3) heterogeneity representation in subgrid scale. Future development is object-oriented to connect them for purposes of simulating the cold regions hydrological cycle over small to medium sized basins by user demand. Representations of hydrological and biogeochemical processes require to be improved to provide new functionality into purpose-built models that will expand the watershed simulation domain while we wait for scientific advancement of numerical algorithms and process understanding.

The lateral fluxes are one main features in watersheds. A way of defining the flux terms of mass, energy, and momentum should be conserved at the boundaries of each discrete element, together with how those fluxes depend on the soil, vegetation and topology [138]. However, there is no horizontal transport between the grid cells in the VIC model [42] and between HRUs in the SWAT [55]. Thus, water, nutrient and pollutant transport

between HRUs within a sub-basin are not considered within a subbasin [139]. This is the major weakness of these widely used watershed models. Thus, water fluxes, including snowmelt water, are not based on those closures of water balance equations but calibration using monitoring data. Bicknell et al. [18] proposed the network of nodes to increase the spatial resolution in the HSPF model. With the advent of numerical algorithms, the use of unstructured grids has been employed widely in computational fluid dynamics [140,141]. This makes it possible to build these conservations of the relevant storages or fluxes between irregular discrete elements, such as HRUs. It will provide the possibility of physical based conservations between element storages and boundary fluxes. Thus, different sources of runoff, snowmelt flow peak, and pollutants can also be formulized in the element. These required sometimes significant rewriting of the model infrastructure, which can be the main direction for model enhancement and improvement [21].

Heterogeneity representation in a grid element is a problem in capturing fine-scale variation of topography, land uses and near-surface soils [12]. For example, the VIC uses statistical distributions to represent each land use in each element and its fraction of the element covered by that particular land use (e.g., forest, grassland, arable, etc.) [42]. In the SWAT, the riparian buffer zones, wetlands, snowpack and other topologies cannot also be resolved explicitly by spatial hydraulic elements, such as HRUs. Heterogeneity of hydraulic properties in each HRU is handled using slope, soil properties and land uses. The lack of spatial detail in spatial elements is a main difficulty to simulate land-atmosphere interaction with agricultural yields, riparian buffer zones and wetlands. These require highly spatial resolution land covers at both the landscape and HRU scale to correctly resolve slope and riparian shape, and crop types. Progress of advanced remote sensing technology may provide higher resolution elevation and land uses to resolve land covers, such as buffer and wetland zone. The ability to resolve land covers with more spatial resolution within a HRU would provide improvements over the current watershed models simulation. This can potentially prove useful for improving current approaches used in these watershed models.

In cold regions, there are knowledge gaps to quantify trends and processes of snow and ice. It is difficult to efficiently obtain and use these data, such as risks and cascading impacts, when simulating large, remote, heterogeneous areas in cold regions [12]. Representation of many hydrological and biogeochemical processes are oversimplified, such as hysteretic nature of soil saturation and unsaturation, and microbially mediated nitrification and denitrification [26,50,51]. Watershed models like SWAT and VIC have a potential for improvement by incorporating more realistic, process-oriented parameterizations for simulating biogeochemical processes. For example, the VIC was coupled with earth system models [21,142,143] and in incorporating microbial-mediated redox reaction into SWAT [50,51]. Improved description of the agroecosystem and biogeochemical processes required to be added. For example, the plant growth module of SWAT is from the Environmental Policy Integrated Climate (EPIC) model [56]. Biomass growth is calculated using daily accumulated heat units under micronutrient response. The growth of different plant species is regulated by plant attributes in the plant database. In the VIC, plant phenology is not dynamic and photosynthesis, autotrophic respiration, and heterotrophic respiration are simulated using a simplified function [144]. Biomass growth is not simulated. Therefore, a model should also be capable of simulating multicultural plant communities, including different types of crop growth and yields as well as greenhouse gas emissions. Numerous types of crops require to be parameterized as the general crop model. The standard library with the parameterization facility should be developed [55]. As the watershed models, these watershed models have been limited in process descriptions for phenomena of different crops with related agricultural management, such as grazing, manuring and fertilization. Furthermore, the evapotranspiration and the complex feedback between the atmosphere and the terrestrial hydrological cycle can be improved due to improvement of plant growth modeling and agricultural management, such as fertilization and manure applications [145,146].

Interaction between groundwater and surface water is key in simulating wetlands and groundwater storage [147]. Groundwater and surface water models in the SWAT and VIC were not designed to account for the interaction between saturated and unsaturated water movement. For example, in groundwater submodel, it is important to address the variable specific yield above the water table, as the specific yield depends on the actual soil moisture profile and availability of that water and lateral groundwater flow to recharge streamflow [26,128,148]. Soil water vertical transport in the shallow aquifer is controlled by three processes: evapotranspiration, infiltration and percolation to a deep aquifer, and horizonal baseflow to its neighbor elements. The coupling of unsaturated zone and saturated zone, which is not represented well in the conceptual models, efficiently simulates the balance between the surface water and groundwater on a physical basis. This can allow to simulate more complicated hydrological processes, such as water table dynamics, and soil water tension.

## 6. Concluding Remarks

Watersheds in cold regions provide water, food, biodiversity and ecosystem service. However, the increasing demand for water resources and foods and climate change challenges the environmental sustainability and biodiversity. Particularly, watersheds in cold regions are more sensitive to changing climate due to their glaciers retreat, and permafrost. Our ability to understand and describe the underlying watershed processes are still limited. Integrated plan and management can support and enhance multiple objectives, such as water availability, access, and soil health across all scales, specially at dealing with impacts from changes in the cryosphere for energy, agriculture, and ecosystems, even some controversial water supply for natural protection and industrial development. This requires not only changes of our management structures, but new and more sophisticated tools. Understanding interactions among hydrological and biogeochemical processes through soil and landscapes is central to research in the fields of soil genesis, soil physics, and hydrology and modeling is unique possibility to integrate these complex hydrological and biogeochemical processes. Therefore, integrated watershed modeling is critical to sustainable water and soil resource management in cold regions. Most of widely used watershed models are only approximations to the real-world complexity of hydrological and biogeochemical processes. It is also obvious that the epistemic issues with hydrological data mean that we would not expect even a perfect model to provide perfect predictions. Watershed models need to be improved in two main areas: (1) heterogeneity representation of hydrological and biogeochemical processes in subgrid scale, and (2) lateral transport of water and nutrient between grids. Due to process simplification, empirical representation and sub-grided heterogeneity, the existing watershed models need to be calibrated and validated against observed data. Therefore, acquisition of new data and new experimental work is still helpful not only for model development but for more accuracy assessment. With advent of computer power and numerical algorithms, watershed models can be improved by using more sophisticated representations. This will provide new functions and increasing use in the water resources planning, development, assessment, and management.

**Author Contributions:** Conceptualization, J.W.; methodology, J.W.; writing—original draft preparation, J.W., N.K.S., M.A.D., T.W.M. and S.N.B.; writing—review and editing, J.W., N.K.S., M.A.D., T.W.M. and S.N.B.; visualization, J.W.; project administration, J.W.; funding acquisition, J.W. All authors have read and agreed to the published version of the manuscript.

**Funding:** This research was funded by Alberta Economic Development and Trade via Campus Alberta Innovation Program (CAIP) Research Chair (Grant No. RCP-12-001-BCAIP).

**Informed Consent Statement:** Not applicable.

**Data Availability Statement:** Not applicable.

**Conflicts of Interest:** The authors declare no conflict of interest.

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
