# Peer review of "Modelling Watershed and River Basin Processes in Cold Climate Regions: A Review"

_water, doi:10.3390/w13040518_

Round 1

Reviewer 1 Report

The manuscript is a review on the modelling of processes at watersheds and river basins scale in cold climate regions. The paper is interesting and, in my opinion there are only a few drawbacks in the paper, which can be eliminated by carrying out some minor revisions following the list of comments below.

I suggest to modify the title as follows: “Modelling watersheds and river basins processes in cold climate regions: A review”.

The article is very long so please avoid the repetition of similar sentences in the text, e.g. Lines 40-42 and lines 45-47.

Is it possible to add for each subsection (from 4.1 to 4.4) a table summarizing the applications? For instance it could be useful a table with 5 columns: authors, case study, variables, model and results.

Check the reference at lines 241-241, the number is missing.

The English should be slightly improved.

Author Response

Reply to Reviewer #1

(1) The manuscript is a review on the modelling of processes at watersheds and river basins scale in cold climate regions. The paper is interesting and, in my opinion there are only a few drawbacks in the paper, which can be eliminated by carrying out some minor revisions following the list of comments below.

Re: Thank you so much for your valuable comments for improving the paper.

(2) I suggest to modify the title as follows: “Modelling watersheds and river basins processes in cold climate regions: A review”.

Re: We agree with the suggestion and the title has been modified accordingly.

(3) The article is very long so please avoid the repetition of similar sentences in the text, e.g. Lines 40-42 and lines 45-47.

Re: Following the suggestion, we have rephrased the stated Lines.

(4) Is it possible to add for each subsection (from 4.1 to 4.4), a table summarizing the applications? For instance, it could be useful a table with 5 columns: authors, case study, variables, model and results.

Re: We understand that a summary table would be great for readers. However, it is difficult to list all studies one by one in a single table or multiple tables. Instead, it should be better to discuss several main issues in different sections.

(5) Check the reference at lines 241-241, the number is missing.

Re: Thank you so much for pointing this out. We have now carefully checked the references and used EndNote, a referencing software.

(6) The English should be slightly improved.

Re: The manuscript has now undergone through English proof reading. We hope the readability has improved.

Reviewer 2 Report

The paper presents a review of watershed modelling and characteristics of watersheds in cold regions, evidences of their impacts of cold processes on hydrological and biogeochemical processes and ecosystems, and previous studies related to watershed modelling and applications in clod regions. The paper further identifies knowledge gaps in modelling river basins and research priorities in future model development. The review would be interesting to readers of this journal. I recommend to accept it with some major revisions as suggested in the following comments.

General comments:

1. Lack of a linkage or connection between review of current status of watershed modeling and applications in Section 4 and issues identified in Section 5, such as knowledge gaps and problems related to watershed modeling (e.g. processes across a wide range of scales, absence of lateral fluxes, poor representations of hydrologic and biogeochemical processes, and inability to capture the heterogeneity).

For most parts of Section 4, it just simply lists the results of each of studies and conclusions. It would be much better if authors can provide comments on (1) what is its success (or advancement) to address issues related to modeling in cold region, (2) what scale was implemented if the detailed processes was incorporated, (3) what processes models that improve representation of various processes that are critical in a cold region but have not been incorporated in watershed models, (4) what is failure that key processes, features, or heterogeneity cannot be captured by the model, and (5) what are knowledge gaps in cold region. These will help make connection to Section 5. Section 5 might focus on how to address the issues/problems identified in Section 4.

2. In Section 5, the identified future priorities in watershed modeling are generally not specific for the cold region. For examples, authors identified three main problems related to modeling (1st para., page 11): lateral fluxes, representations of hydrologic and biogeochemical processes, and heterogeneity. These are common problems for watershed modeling in all regions. It would be much interesting to the readers to identify specific areas related the clod regions.

3. In Section 5, only one issue identified was related to the cold region: knowledge gaps in the distribution and characteristics of cryospheric variables in the extent and ice content of permafrost in mountains, current glacier ice volumes, trends in lake and river ice, and the spatial and temporal variation of snow cover (last para, page 10). Authors do not identify what might be research directions and/or priorities to address them, what have been done, and what might be promising.

More specific comments:

4. Lines 76-77, there is gap from a sensitive nature of a cold region to climate change, importance of process integration in watershed, watershed modeling to needs for an interdisciplinary review of watershed modelling. The gap might be key issues of watershed modeling for cold regions.

5. Lines 92-93, “water budget in a watershed is balanced by precipitation, evapotranspiration, infiltration, and runoff.” For a cold region, glacier, snowpack, permafrost, etc. (frozen water components) are also important parts of water budget. Some of models discussed in this paper may not have a good process to capture those budget.

6. Lines 187-188, “…in two watersheds in eastern Canada (Bras d’Henri watershed and Bras d’Henri watershed).” What is the name for the other watershed?

7. Line 196, typo: “When temperature is lower than 0oC, soil water thaws” – it should be “higher than”.

8. Section 4. I would suggest that authors include some process models that are critical for a cold region from the previous studies. The reviews limited to SWAT, VIC, CRHM, and HSPF may miss some models for cold-region specific processes. These individual cold process-specific models could be a future research priority to incorporate them into watershed models. For example, snowpack and melting process using energy balance model (e.g. Utah Energy Balance model, Tarboton et al 1994, 1996; and others); Algorithms developed for near-surface soil freeze-thaw cycle for contiguous US, Zhnag et al 2003 JGR; The process implemented in CLM 4.5 for simulating near-surface soil freeze-thaw cycle with leveraging satellite monitoring data, Guo et al 2018 JGR atmospheres. There could be more examples that demonstrate future potential approach to incorporate better representations of cold processes into watershed models.

9. Section 5. Authors identified one of three key problems is heterogeneity of key parameters (or physical properties). SWAT and HSPF use HRU as a computational unit with a uniform property for slope, soil type, and landuse. In theory, these properties can be further discretized by using smaller HRUs. The limitation might be computation power and future approach is to re-write codes to make it parallel. But more important to the cold region is to discretize HRUs to temperature dependent unit in addition to the properties of slope, soil, and landuse and location specifics. The current HRUs has no specific locations within the subbasin, which cannot fully capture temperature driven activities (snowmelt, freeze-thaw) at different elevation within a watershed.

10. Section 5. For cold region, site access is typically remote and extreme cold can limit in situ monitoring devices. In these circumstances, the fusion of sparsely available data with hydrological models and/or leveraging satellite data is an effective method to improve process-based watershed modeling in cold region, specifically quantify water fluxes and storage dynamics at different temporal and spatial scales. Author might explore more on approaches that address unique issues related to the cold region instead of common modeling issues applicable to everywhere.

Author Response

Reply to Reviewer #2

The paper presents a review of watershed modelling and characteristics of watersheds in cold regions, evidences of their impacts of cold processes on hydrological and biogeochemical processes and ecosystems, and previous studies related to watershed modelling and applications in clod regions. The paper further identifies knowledge gaps in modelling river basins and research priorities in future model development. The review would be interesting to readers of this journal. I recommend to accept it with some major revisions as suggested in the following comments.

Re: We would like to thank the reviewer for your valuable comments for improving the paper.

General comments:

(1) Lack of a linkage or connection between review of current status of watershed modeling and applications in Section 4 and issues identified in Section 5, such as knowledge gaps and problems related to watershed modeling (e.g. processes across a wide range of scales, absence of lateral fluxes, poor representations of hydrologic and biogeochemical processes, and inability to capture the heterogeneity).

Re: While we understand the concern raised, we would like to clarify the following. The knowledge gaps in Section 5 are because of some fundamental problems in the existing watershed model structures and process representations. For example, the issues of the lateral fluxes and heterogeneity can be attributed to model structure assumptions of HRUs and high-resolution data unavailability. In contrast, the poor representation of hydrological and biogeochemical processes, including snowpack processes, are because of poor understanding of these processes or mathematical difficulties.  However, these fundamental issues have not been presented in Section 2. Therefore, these are fundamental problems in watershed modelling that should be highlighted in this section. Of course, these main challenges should be the directions of the future research. Section 4 is to present some applications and demonstrates the successes of these watershed models what have been achieved. Therefore, this section does not repeat the model structure and their process simplification in Section 2 and discusses main challenges in Section 5. We clarify these further in the revised version in Section 5.

(2) For most parts of Section 4, it just simply lists the results of each of studies and conclusions. It would be much better if authors can provide comments on (1) what is its success (or advancement) to address issues related to modeling in cold region, (2) what scale was implemented if the detailed processes was incorporated, (3) what processes models that improve representation of various processes that are critical in a cold region but have not been incorporated in watershed models, (4) what is failure that key processes, features, or heterogeneity cannot be captured by the model, and (5) what are knowledge gaps in cold region. These will help make connection to Section 5. Section 5 might focus on how to address the issues/problems identified in Section 4.

Re: We really appreciate the insights presented. It is difficult to list successes and failures of all the studies one by one because of page limitation. Instead, we demonstrate their successful applications of these watershed models in this section. Failures or problems are presented in Section 5. In practice, due to issues with model structures and representations of processes, snow and ice processes cannot be resolved well in the watershed models. We added a subsection 3.4 to discuss the issue explicitly. This is a key knowledge gap in cold regions. In this revised version, we added some further description of issues of snow processes in Section 5 to link with Section 2.

(3) In Section 5, the identified future priorities in watershed modeling are generally not specific for the cold region. For examples, authors identified three main problems related to modeling (1st para., page 11): lateral fluxes, representations of hydrologic and biogeochemical processes, and heterogeneity. These are common problems for watershed modeling in all regions. It would be much interesting to the readers to identify specific areas related the clod regions.

Re: We totally agree with the comment, and we clarify it in the revised version. In practice, the specific issues related to the cold regions are snowpack and ice processes. Water and heat processes in snowpack or soil ices are one-dimensional model, based on HRUs grid. Thus, lateral fluxes and its heterogeneity are fundamental issues of model structures related to snowmelt water and permafrost. Particularly, there are no information on exchanges or fluxes between neighbour HRUs. This could be improved if high resolution data are available in the future but should be indicated in future research. We also modified Table 1 to show snowpack models related to the cold regions.    

(4) In Section 5, only one issue identified was related to the cold region: knowledge gaps in the distribution and characteristics of cryospheric variables in the extent and ice content of permafrost in mountains, current glacier ice volumes, trends in lake and river ice, and the spatial and temporal variation of snow cover (last para, page 10). Authors do not identify what might be research directions and/or priorities to address them, what have been done, and what might be promising.

Re: Thank you for your suggestions. We have elaborated it by adding: “Particularly, water and heat transport in snowpack layers are a three-dimensional process. In these widely used watershed models, both one-layer and two-layer models consider only vertical water transport and snow accumulation.” Like lateral flow issue, the snowmelt water are main sources of flooding in Spring in cold regions. Freeze-thaw are one of main processes in cold regions. Water and heat transport in freeze-thaw processes are still a knowledge gaps in watershed modelling.

More specific comments:

(5) Lines 76-77, there is gap from a sensitive nature of a cold region to climate change, importance of process integration in watershed, watershed modeling to needs for an interdisciplinary review of watershed modelling. The gap might be key issues of watershed modeling for cold regions.

Re: We agree, and we have rephrased the indicated sentences.

(6) Lines 92-93, “water budget in a watershed is balanced by precipitation, evapotranspiration, infiltration, and runoff.” For a cold region, glacier, snowpack, permafrost, etc. (frozen water components) are also important parts of water budget. Some of models discussed in this paper may not have a good process to capture those budget.

Re: We agree, and we have rephrased the indicated sentences.

(7) Lines 187-188, “…in two watersheds in eastern Canada (Bras d’Henri watershed and Bras d’Henri watershed).” What is the name for the other watershed?

Re: These are not watersheds but samples from 58 lakes in the Tibetan Plateau

(8) Line 196, typo: “When temperature is lower than 0oC, soil water thaws” – it should be “higher than”.

Re: Thank you so much for pointing this out, it has been corrected in the revised version.

(9) Section 4. I would suggest that authors include some process models that are critical for a cold region from the previous studies. The reviews limited to SWAT, VIC, CRHM, and HSPF may miss some models for cold-region specific processes. These individual cold process-specific models could be a future research priority to incorporate them into watershed models. For example, snowpack and melting process using energy balance model (e.g. Utah Energy Balance model, Tarboton et al 1994, 1996; and others); Algorithms developed for near-surface soil freeze-thaw cycle for contiguous US, Zhnag et al 2003 JGR; The process implemented in CLM 4.5 for simulating near-surface soil freeze-thaw cycle with leveraging satellite monitoring data, Guo et al 2018 JGR atmospheres. There could be more examples that demonstrate future potential approach to incorporate better representations of cold processes into watershed models.

Re: We agree, and we added a subsection 3.4 to briefly review the cold processes, such as snowpack and freeze-thaw. These models have been cited. Section 4 presented some applications. Therefore, it is better that the cold processes modelling is included in Section 3 as part of the characteristics of watershed processes in cold regions.  

(10) Section 5. Authors identified one of three key problems is heterogeneity of key parameters (or physical properties). SWAT and HSPF use HRU as a computational unit with a uniform property for slope, soil type, and landuse. In theory, these properties can be further discretized by using smaller HRUs. The limitation might be computation power and future approach is to re-write codes to make it parallel. But more important to the cold region is to discretize HRUs to temperature dependent unit in addition to the properties of slope, soil, and landuse and location specifics. The current HRUs has no specific locations within the subbasin, which cannot fully capture temperature driven activities (snowmelt, freeze-thaw) at different elevation within a watershed.

Re: We agree with you that HRUs can be further downscaled spatially. Computing power is a limitation, but high-resolution input data is not available in most of the cases. It is true that HRUs have not considered temperature. However, weather drivers, such as temperature, rainfall and radiation, are input parameters in the SWAT, VIC and HSPF. Therefore, this is reasonable not to consider them in HRUs.

(11) Section 5. For cold region, site access is typically remote and extreme cold can limit in situ monitoring devices. In these circumstances, the fusion of sparsely available data with hydrological models and/or leveraging satellite data is an effective method to improve process-based watershed modeling in cold region, specifically quantify water fluxes and storage dynamics at different temporal and spatial scales. Author might explore more on approaches that address unique issues related to the cold region instead of common modeling issues applicable to everywhere.

Re: Thank you for your suggestions. We added a sentence to further clarify: “Particularly, water and heat transport in snowpack layers are a three-dimensional process. In these widely used watershed models, both one-layer and two-layer models consider only vertical water transport and snow accumulation.” Like lateral flow issue, the snowmelt water are main sources of flooding in Spring in cold regions.  

Reviewer 3 Report

No additional comments.

Author Response

Re: Thank you so much for your recommendation.